# Optimizing availability of obstetric surgical care in India: A cost-effectiveness analysis examining rates and access to Cesarean sections

Lina Roa[1]*, Luke Caddell[2], Namit Choksi[3,4], Shylaja Devi[5], Jordan Pyda[6], Adeline A. Boatin[7,8], Mark Shrime[7,9]

1 Department of Obstetrics & Gynecology, University of Alberta, Edmonton, AB, Canada, 2 Department of General Surgery, Stanford University, Stanford, CA, United States of America, 3 Smt. Kashibai Navale Medical College and General Hospital, Pune, Maharashtra, India, 4 School of Healthcare, Rishihood University, Sonepat, Haryana, India, 5 Gudalur Adivasi Hospital, Gudalur, Nilgiris, Tamil Nadu, India, 6 Department of Surgery, Beth Israel Deaconess Medical Center, Boston, MA, United States of America, 7 Program in Global Surgery and Social Change, Department of Global Health and Social Medicine, Harvard Medical School, Boston, MA, United States of America, 8 Department of Obstetrics & Gynecology, Massachusetts General Hospital, Boston, MA, United States of America, 9 Institute of Global Surgery, Royal College of Surgeons in Ireland, Dublin, Ireland

* lroa123@gmail.com

**Data Availability Statement:** All the data used to create the model was extracted from the literature and is available in the manuscript.

## Abstract

The objective of this study is to assess the cost-effectiveness of three different strategies with different availabilities of cesarean sections (CS). The setting was rural and urban areas of India with varying rates of CS and access to comprehensive emergency obstetric care (CEmOC) for women of reproductive age in India. Three strategies with different access to CEmOC and CS rates were evaluated: (A) India's national average (50.2% access, 17.2% CS rate), (B) rural areas (47.2% access, 12.8% CS rate) and (C) urban areas (55.7% access, 28.2% CS rate). We performed a first-order Monte Carlo simulation using a 1-year cycle time and 34-year time horizon. All inputs were derived from literature. A societal perspective was utilized with a willingness-to-pay threshold of $1,940. The outcome measures were costs and quality-adjusted life years were used to calculate the incremental cost-effectiveness ratio (ICER). Maternal and neonatal outcomes were calculated. Strategy C with the highest access to CEmOC despite the highest CS rate was cost-effective, with an ICER of 354.90. Two-way sensitivity analysis demonstrated this was driven by increased access to CEmOC. The highest CS rate strategy had the highest number of previa, accreta and ICU admissions. The strategy with the lowest access to CEmOC had the highest number of fistulae, uterine rupture, and stillbirths. In conclusion, morbidity and mortality result from lack of access to CEmOC and overuse of CS. While interventions are needed to address both, increasing access to surgical obstetric care drives cost-effectiveness and is paramount to optimize outcomes.

**Funding:** MGS receives funding from the Iris O'Brien Foundation. AAB is supported by career development awards from the Eunice Kennedy Schriver National Institute of Child Health and Human Development (K23 HD097300-01) and Massachusetts General Hospital Executive Committee on Research through the Center for Diversity and Inclusion. The funders did not have input on research design or interpretation.

**Competing interests:** I have read the journal's policy and the authors of this manuscript have the following competing interests: MGS serves on the board for Pharos Global Health Advisors, has received speaking fees from Meditech and the American Hospital Association. AAB serves on the scientific advisory board for RHIA ventures and has provided consulting services to DARE Bioscience.

## Introduction

A cesarean section (CS) is a life-saving intervention, and a cost-effective intervention [1–4]. However, CS, particularly those not medically indicated, can expose women to adverse outcomes including hemorrhage, infection, thromboembolism, complications from anesthesia, neonatal respiratory distress, and more [5, 6]. Additionally, with each additional CS there is an increased risk of placenta previa, placenta accreta, and the consequent hemorrhage, peripartum hysterectomy, ICU admission and death [7, 8]. Thus, both underuse and overuse of CS can result in avertable morbidity and mortality.

Globally, CS rates continue to rise. In 2015, 21.1% of all live births were born by CS, nearly double the rate from 2000 [9–13]. However, significant variation in CS rates exists between and within countries [12, 14]. In India there has been a large increase in institutional deliveries and the national CS rate has almost doubled in the past decade from 9% in 2006 to 17% in 2016 [15, 16]. Data from 2015–2016 shows that the urban CS rate [28.2%] is more than double the rural rate [12.9%], and significantly above population-based rates where no mortality benefit has been demonstrated [1, 17]. There is also significant variation in CS rates from 5.8% in the state of Nagaland to 58% in Telangana [16, 18]. This raises the concern that some women in India are unnecessarily exposed to the risks of a non-indicated surgical procedure, while others lack access to surgical obstetric care [16, 18–20]. This variability may be due to differences in the facilities women access, ranging from facilities that do not provide comprehensive emergency obstetric care (CEmOC), which by definition include CS, to those in which non-indicated CS are overused [16, 19, 20].

Optimizing CS availability thus requires a fine balance between ensuring access CEmOC facilities for all women while preventing overuse of CS without medical indication. We evaluated three hypothetical health policy strategies that assume the ability to provide equitable obstetric surgical care to the entire population of India under different assumptions. Although the strategies are hypothetical, the assumptions are based on existing national data for all of India. In Strategy A, we test a health policy strategy where the assumption is that the entire population is given access to CEoMC facilities at the current national access rate and has a CS rate equivalent to the current national average. In Strategy B, the entire population is given access to CEoMC facilities at the current rural access rate and has a CS rate equivalent to the current rural average. In Strategy C, the entire population is given access to CEoMC facilities at the current urban access rate and has a CS rate equivalent to the current urban average. Cost-effectiveness analyses (CEA) compare the costs and outcomes of different strategies and could be used to inform policies on how to scale up obstetric surgical care. The purpose of this study was to assess the cost-effectiveness of the above strategies with different rates of access to CEmOC facilities and CS rates.

## Methods

### Overview

We designed a state-transition microsimulation to simulate the natural history of pregnancy and relevant comorbidities. The model incorporated data from urban and rural India on access to skilled birth attendance, access to CEmOC, and CS rates. Model outcomes included clinical events, maternal morbidity, maternal mortality, neonatal mortality, and economic costs. We evaluated alternative approaches to CS in settings in India that differ in access to CEmOC care and CS rates.

### Model

A decision analysis model was built to simulate the natural history of pregnancy and associated complications. We considered all women of reproductive age (14–49 years) in India. The time

horizon included a woman's reproductive years of 35 years (49–14 = 35 years). A Markov model was built with four health states: pregnant (utility = 0.99), non-pregnant(utility = 1), history of hysterectomy (0.94), dead (utility = 0). During each year women transition from one health state to another. We performed a first order Monte Carlo Simulation using a one-year cycle, with 10,000 simulations run. On entry into the model and in each cycle, women face a probability of becoming pregnant based on India's fertility rate. Once pregnant, women have a probability of having access to CEmOC (Fig A in S1 Text). Depending on a woman's history of previous CS, she will have different probabilities of developing placenta previa and placenta accreta (Table A in S1 Text). Similarly, she will have different probabilities of having a non-indicated ante-partum CS, a non-indicated intra-partum CS, a trial of labor after CS [TOLAC] and vaginal birth after CS (VBAC). Women undergoing vaginal delivery and CS have differing probabilities of complications of pregnancy (post-partum hemorrhage, peri-partum hysterectomy, ICU admission). For women that do not have access to surgical care, a proportion develop complications (obstructed labor, fistula, incontinence, stroke, hemorrhage). Trackers were used to estimate the number of complications in each strategy. Mortality was conditional on complication severity (e.g. complications requiring ICU admission translated to higher probability of mortality) and underlying morbidity (e.g. patients with placenta previa had higher post-partum hemorrhage rates than patients without it).

Strategies evaluated included different availability to CS depending on a combination of access to facilities that provide CS and CS rates (Fig 1). The United Nations Population Fund defines CEmOC as care typically delivered in hospitals that includes basic obstetric care plus

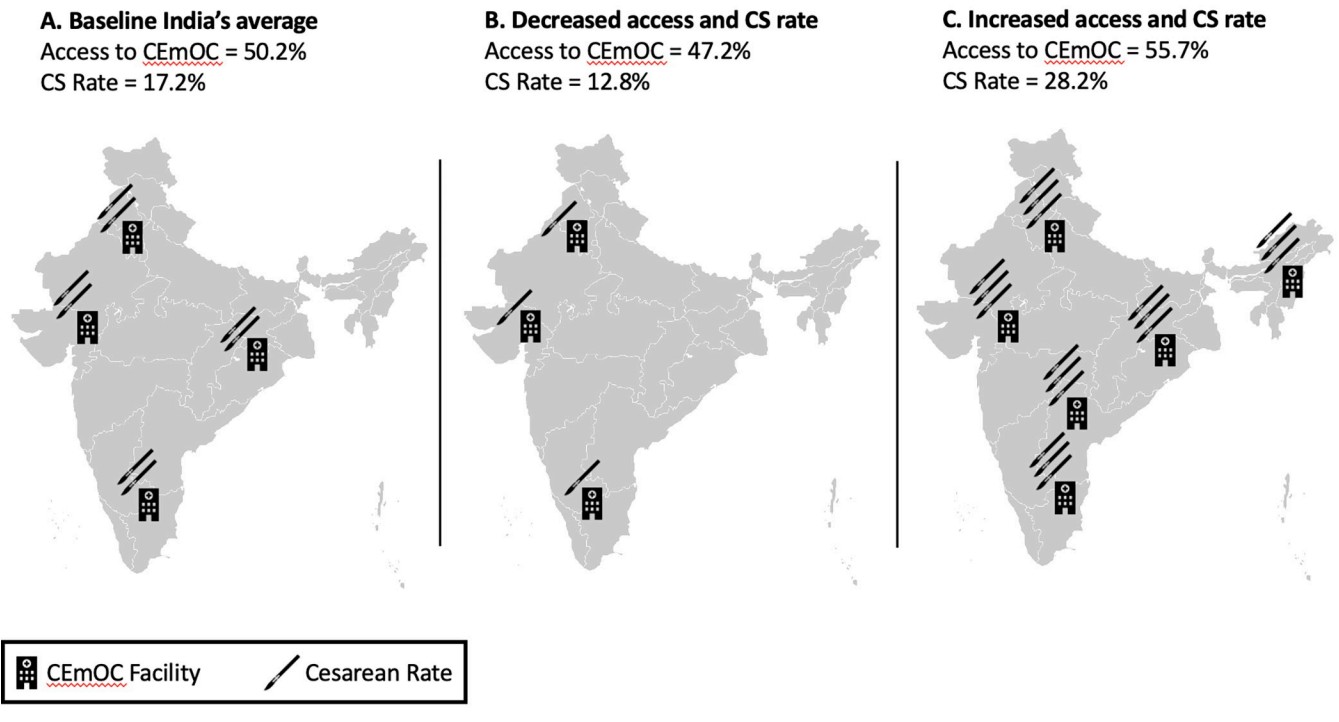

**A. Baseline India's average**
Access to CEmOC = 50.2%
CS Rate = 17.2%

**B. Decreased access and CS rate**
Access to CEmOC = 47.2%
CS Rate = 12.8%

**C. Increased access and CS rate**
Access to CEmOC = 55.7%
CS Rate = 28.2%

🏥 CEmOC Facility        ╱ Cesarean Rate

**Fig 1. Strategies to scale up CS care.** Graphic representation of the three strategies: [A] is the baseline strategy with India's national average access to CEmOC and average CS rate; [B] is the strategy that reflects India's current access to CEmOC and CS rates in rural areas, which are poor and adequate, respectively; [C] is the strategy that reflects India's current CEmOC access and CS rates in urban areas, which have better access to CEmOC than the average and a higher CS rate than the rest of India. The facilities icon represent access to CEmOC, and the scalpels represents CS rate. The icons are a visual representation and do not illustrate proportional changes between the strategies. Base Layer from: https://commons.m.wikimedia.org/wiki/File:BlankMap-India2.92.png.

capabilities to perform cesarean sections, blood transfusions and resuscitation of newborns. India has extensive data on access to deliveries at health care facilities but robust data on CEmOC facilities is limited to one state. Therefore access to CEmOC was defined as the probability of accessing a facility that provides an institutional delivery times the probability of accessing a facility that provides CEmOC. Data on access to institutional deliveries was obtained from national data on rural and urban areas while data on access to CEmOC facilities was available only from the state of Gujarat. These three strategies are: (A) The population has access to CEmOC and CS rates that reflect India's current national average access and rates (49.58% access to CEmOC and 17.2% CS rate). This strategy would represent the elimination of current urban-rural inequalities in access and CS rates. (B) A strategy in which the population's access to CEmOC and CS rates reflects India's current access and rates in rural areas (47.19% access to CEmOC facilities and 12.8% CS rate). This strategy would represent the decrease of India's average CS rate, with a decrease in average access to CEmOC. (C) A strategy in which the population's access to CEmOC and CS rates reflect India's current access and rates in urban India (55.74% access to CEmOC facilities and 28.2% CS rate). This strategy would represent an increase in access to CEmOC, with an increase in CS rates from the average. We were limited to using strategies for which outcome data exists and therefore focused on currently available data from the national, rural, urban settings and did not simulate other scenarios with different CS rates or CEmOC access. We used Monte Carlo microsimulation to generate the number of events per woman such as pregnancies, CS, live births, complications, etc. The model was created using Amua version 0.2.7 [21].

## Data

Parameters and assumptions used in the model are presented in Table 1 and S1 Text. Event probabilities, including pregnancy related complications, where derived from the literature. Preference was given to literature from India, when this was not available, data from low-and-middle income countries (LMIC), with a preference for South Asia were used. As a last resort, data from HIC studies were utilized. Data on facility-based deliveries, rural and urban population, and CS rates were obtained from India's national surveys [16]. Beta distributions were used to model probabilities. Dirichlet distributions were generated for probabilities that were standardized. Uniform distributions were used for health utility values. Half cycle corrections for utilities were incorporated. Non-age weighted utility weights were obtained from the 2015 Global Burden of Disease study [22]. If these were not available, they were obtained from existing literature. The willingness to pay (WTP) threshold is an estimate of what a consumer of health care is prepared to pay for the given health benefit and it is often the country's GDP per capita. The WTP threshold we used was $1,939.61 USD, India's GDP per capita in 2017 [23].

## Costs

Selected costs in USD are provided in Table 1. Costs for cesarean section and vaginal delivery were obtained from studies that used a societal perspective. Direct health costs were included the fixed salaries of healthcare providers, variable facility, and drug charges for transfusion, antibiotics, cesarean section, hysterectomy, anesthesia, and ICU-level care [24–28]. Transportation to higher levels of care and to and from treatment facility were included and indirect costs of the patient time in seeking care were included when available [24–28]. Where multiple cost figures were available in the literature, mean costs were used and gamma distributions were calculated. We assessed the validity of the costs obtained from the literature by checking with providers in India familiar with the costs of obstetric care. Based on the literature, we established a range for sensitivity analysis. Half cycle corrections were incorporated. All costs

**Table 1. Input parameters and model assumptions.**

| Variable | Baseline [probability] | Beta Distribution | Reference |
|---|---|---|---|
| Probability a woman gets pregnant in a given year | 0.0654 | [14.882, 212.76] | [41] |
| **Markov States** | | | |
| Probability that a woman has a history of hysterectomy | 0.017 | [5378.14, 310982.86] | [42] |
| Probability that a woman has a history of CS | 0.172 | [42994.32, 206972.68] | [16] |
| Probability that a woman in a rural area has a history of CS | 0.128 | [23020.67, 156828.33] | [16] |
| Probability that a woman in an urban area has a history of CS | 0.282 | [19773.28, 50344.72] | [16] |
| Probability that a woman has no history of CS | Complement | | |
| **Access** | **Probability** | **Beta Distribution** | **Reference** |
| Probability that a woman has access to CEmOC | 0.502 | [125483.43, 124483.57] | [16] |
| Probability that a woman in a rural area has access to CEmOC | 0.472 | [84888.73, 94960] | [16] |
| Probability that a woman in an urban area has access to CEmOC | 0.557 | [39055.73, 31062.27] | [16] |
| **Complications** | **Probability** | **Beta Distribution** | **Reference** |
| Probability of a woman developing an absolute indication for antepartum CS | 0.0362 | [893.49, 23788.51] | [43] |
| Probability of PPH in the setting of vaginal delivery | 0.009 | [1174.32, 129305.68] | [44] |
| Probability of hysterectomy in the setting of vaginal delivery | 0.0003 | [4.39, 14632.61] | [45] |
| Probability of intrapartum CS in the setting of TOLAC | 0.739 | [3346.93, 1182.07] | [43] |
| Probability of not having an indication for an antepartum CS | 0.0094 | [42.98, 4529.02] | [43] |
| Probability of not having an indication for an intrapartum CS in the setting of TOLAC | 0.018 | [21, 1161] | [43] |
| Probability of successful TOLAC | Complement | | |
| Probability of PPH in the setting of placenta accreta | 0.6 | [18, 12] | [46] |
| Probability of hysterectomy in the setting of placenta accreta | 0.765 | [42.84, 13.16] | [47] |
| Probability of PPH in the setting of placenta previa | 0.2111 | [57, 213] | [48] |
| Probability of uterine rupture in the setting of TOLAC | 0.0169 | [25, 1454] | [49] |
| Probability of hysterectomy in the setting of uterine rupture during TOLAC | 0.0819 | [14.61, 163.75] | [40] |
| Probability of ICU admission in the setting of a peri-partum hysterectomy | 0.357 | [19.99, 36.01] | [45] |
| Probability of all-cause for obstetric mortality during ICU admission | 0.2165 | [42, 152] | [50] |
| Probability of a woman developing an absolute indication for intrapartum CS | 0.144 | [3426, 20348] | [43] |
| Probability of mortality in the setting of an absolute indication [placenta accreta, placenta previa] for antepartum CS | 0.00323 | [15.95, 4920.62] | [51–53] |
| Probability of mortality in the setting of an absolute indication [uterine rupture, prolonged obstructed labor] for intrapartum CS | 0.0025 | [15.96, 6367.04] | [44, 54, 55] |
| Probability of mortality in the setting of PPH during vaginal delivery | 0.006 | [11.01, 18618.99] | [56] |
| Probability of emergency hysterectomy during CS | 0.0027 | [39.52, 14597.48] | [45] |
| Probability of PPH in the setting of uterine rupture | 0.314 | [50.55, 110.45] | [57] |
| Probability of mortality in setting of PPH during vaginal delivery without access to care | 0.0006 | [11.01, 18618.99] | [56] |
| Probability of developing a fistula following vaginal delivery in setting of absolute indication for CS | 0.0001 | [16, 159967] | [58] |
| Probability of developing incontinence during vaginal delivery in the setting of absolute indication for CS | 0.0011 | [15.98, 14512.97] | [58] |
| Probability of developing stroke during vaginal delivery in the setting of absolute indication for CS | 0.0015 | [15.97, 10633] | [59] |
| **Costs** | **Cost [US $]** | **Gamma Distribution** | **Reference** |
| Vaginal delivery at a facility | 153.70 | [16, 9.51] | [24, 25] |
| CS | 745.66 | [16, 47.93] | [24, 26, 27] |
| Managing PPH, including blood transfusion | 79.89 | [16, 4.99] | [24] |
| Peripartum hysterectomy | 2241.42 | [16, 140.09] | [26] |

(*Continued*)

**Table 1.** (Continued)

| | | | |
|---|---|---|---|
| ICU admission | 1172.50 | [16, 28.03] | [28, 60] |

**Assumptions:**

Fertility rate is the same throughout India. If a woman has no access to CS, then she also has no access to blood transfusion, hysterectomy, or ICU-level care. The cost of uterine rupture is equivalent to the cost of CS. Absolute indications for both antepartum CS and intrapartum CS are independent and mutually exclusive. CEmOC capability is equivalent across India [data used from Gujarat state]. Access to CEmOC care is equal to the probability that someone delivers in a facility times the probability that the facility has CEmOC capabilities. Neonatal mortality occurs if the woman has an absolute indication for a CS but does not have access to CS. The probability of PPH following TOLAC which ends in an intrapartum CS is equivalent to the probability of PPH after any CS. In rural areas where current CS rate is below the WHO recommended rate, there were no CS performed without indication. If a woman has access to cesarean section, neonatal survival is 100%.

*Complement probabilities were obtained by subtracting from 1 the probabilities of all parallel branches to obtain the complement. Cost of medical services in India are presented in 2016 USD

** CS: Cesarean section; TOLAC: trial of labor after CS; PPH: post-partum hemorrhage; VBAC: vaginal birth after CS; ICU: intensive care unit, CEmOC: comprehensive emergency obstetric care

were standardized by adjusting for inflation by using the World Bank GDP deflators [29]. Costs were converted using the purchasing power parity conversion factor for private consumption provided by the World Bank to obtain 2016 USD [30].

## Analysis

Cost-effectiveness analysis is a policy making tool that provide a framework to assess which strategy maximizes health benefit per dollar spent. The output of a CEA that compares the strategies is the incremental cost-effectiveness ratio (ICER). We calculated the ICER for each of the three strategies, defined as the additional cost of a specific strategy divided by its additional benefit in quality-adjusted life-year (QALY), compared with the next least expensive strategy. Interventions were considered to be cost-effective if the cost-effectiveness ratio was less than the per capita gross domestic product (GDP) of US$1,939 in India in 2017. Mean number of complications per woman, maternal mortality per 100,000 live births, and neonatal mortality per 1,000 live births were calculated from the model output. The number of complications per 10,000 women were calculated and extrapolated to India's 2018 female population of reproductive age [31]. The analysis was conducted using Amua version 0.2.7 [21].

## Sensitivity analysis

All parameters were subject to probabilistic sensitivity analyses. Selected parameters were subject to one-way sensitivity analysis to assess the impact of parameter uncertainty on the model. The following parameters were tested: probability of post-partum hemorrhage (PPH) from a vaginal delivery, placenta previa, placenta accreta and from uterine rupture; probability of intrapartum CS after TOLAC, of uterine rupture, of ICU admission and of hysterectomy; probability of mortality from PPH, from an indicated intra- and ante-partum CS and from ICU admission; cost of ICU, of a CS delivery, and of a vaginal delivery. Access to CEmOC facilities and CS rates were subject to two-way sensitivity analysis.

## Calibration

Data on probability of non-indicated intra-partum and ante-partum CS were not available in the literature and therefore these were estimated from the model. The probability of non-indicated CS for each of the strategies was varied at regular intervals while all other variables remained constant until we obtained the known overall CS rate for each of the strategies, thereby allowing us to estimate the rates of non-indicated CS.

## Results

### Cost-effectiveness of strategies

Strategy C (urban data), which extends the access and rates of CS currently observed in urban India to the entire population and has the greatest access to CEmOC facilities (55.74%) and highest CS rate (28.2%), is a cost-effective strategy. The ICER is 354.90, is the cost-effective strategy that provides the greatest benefit (QALYs) while remaining below the WTP threshold of $1,939.61 (Fig 2). Strategy B (rural data) has an ICER of 5.20 and while cost effective, it provides less benefit (QALYs) than strategy C. Strategy A (national average) is weakly dominated and not in the cost-effectiveness frontier. Both strategies B (rural data) and C (urban data) are associated with increase and decrease of particular number of complications (Table 2). Strategy C (urban data) had more complications related to high CS rates such as higher numbers of placenta previa, placenta accreta, and ICU admissions. However, in this strategy, it was calculated that there would be 860 fewer cases of fistula per year and that neonatal mortality would decrease by 11 neonatal deaths per 1,000 live births compared with the national average strategy. In Strategy B (rural data), despite a decrease in some complications including 2,868 fewer cases of placenta accreta per year and 2,772 fewer ICU admissions for PPH, the neonatal mortality increased by 3 deaths per 100,000 live births compared with Strategy A(national average).

### Sensitivity analysis

A one-way sensitivity analysis was conducted, and the model was not sensitive to any of the parameters tested. A two-way sensitivity analysis of access to CEmOC facility and CS rates was conducted which demonstrated that access to CEmOC drives the cost-effectiveness of the strategy more than access to CS rates (Fig B in S1 Text).

## Discussion

### Main findings

Among the three strategies tested, strategy C (urban data) is cost-effective. Strategy B (rural strategy) is on the cost-effectiveness frontier, but a greater benefit was obtained from Strategy C (urban data). While in strategy C (urban data) CS rates are the highest—and higher than current recommendations—this strategy allowed for the highest access to CS. Two-way sensitivity analysis demonstrated that higher access to CEmOC facilities, is driving the cost-effectiveness as opposed to increases in the CS rate. We simulated the future number of maternal and neonatal complications with each of the current three strategies. The simulation shows an increase in the number of maternal comorbidities in strategy C (urban data) likely driven by high CS rates rather than increased access to surgical care. These complications, are typically associated with increasing CS rates and include higher rates of placenta previa, placenta accreta, PPH, and ICU admission. However, in this strategy [urban data], neonatal mortality and number of fistulas are greatly decreased, likely due to the better access to surgical care in this strategy. In Strategy B (rural data) higher rates of neonatal mortality and stillbirth occur. Interestingly, maternal mortality was highest in Strategy C (urban data), despite this strategy having the highest access to surgical care. We hypothesize this is because 55.7% of the population having access to surgical care is still not enough to manage the complications in the next 35 years from rising CS rates such as placenta accreta, peripartum hysterectomy, and ICU admission. Furthermore, it is likely that women with more co-morbidities and complicated pregnancies get referred to urban centers that as a result perform more CS and have a higher mortality due to their case mix. There was little

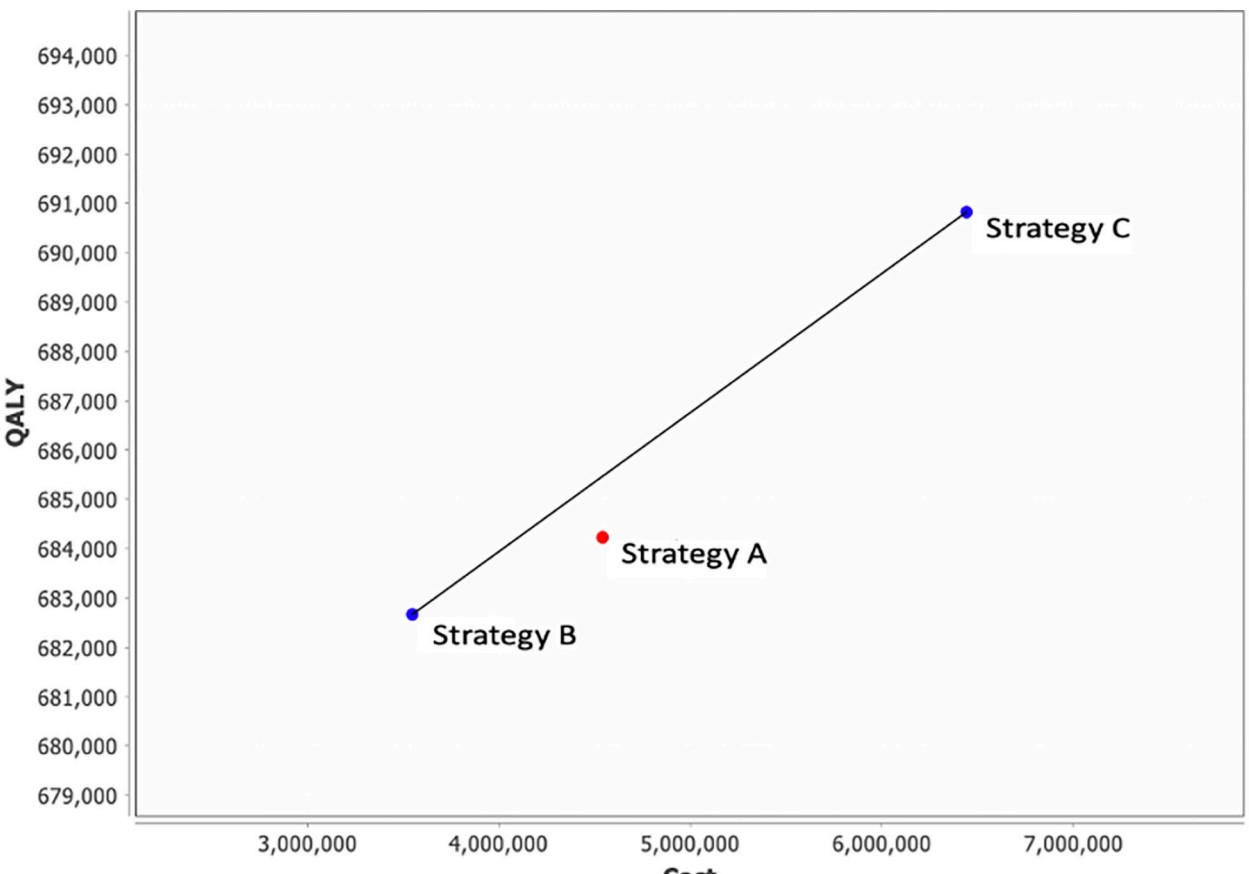

| Strategy | Cost (USD) | QALY | ICER |
|---|---|---|---|
| **B: Data from Rural India**<br>Access to CEmOC: 47.2%<br>CS Rate: 12.8% | 3,547,844 | 682,670 | 5.20 |
| **A: Data from national average**<br>Access to CEmOC: 50.2%<br>CS Rate: 17.2% | 4,541,842 | 684,229 | Dominated |
| **C: Data from Urban India**<br>Access to CEmOC: 55.7%<br>CS Rate: 28.2% | 6,442,120 | 690,825 | 354.90 |

**Fig 2. Cost and QALY for each of the strategies to scale up CS care.** Plot of cost [USD] and utility [QALY] for the three strategies modelled and corresponding costs, quality-adjusted life years [QALY], and incremental cost-effectiveness ratio [ICER]. The strategies reflect equitable provision of cesarean sections for all of India based on existing national data [Strategy A], rural data [Strategy B] and urban data [Strategy C].

**Table 2. Events per 10,000 women during their reproductive lifetime and estimates for the entire female population in India.**

| Strategies | Strategy A [National] | | Strategy B [Rural] | | Strategy C [Urban] | |
|---|---|---|---|---|---|---|
| Access to CEmOC [%] | 50.2 | | 47.2 | | 55.7 | |
| CS rate [%] | 17.2 | | 12.8 | | 28.2 | |
| Differences between strategies | Strategy A [Baseline] | | Change from Strategy A to B [B-A] | | Change from Strategy A to C [C-A] | |
| | Per 10,000 women | Per total population | Per 10,000 women | Per total population | Per 10,000 women | Per total population |
| Pregnancies to full term | 646 | 20,998,079 | +2.1 | +68,735 | -0.2 | -7,743 |
| Vaginal Deliveries | 522 | 16,957,079 | +47 | +1,539,119 | -84 | -2,732,176 |
| Cesarean Sections | 173 | 5,628,779 | -58 | -1,888,050 | +116 | +3,764,628 |
| Neonates that survive | 621 | 20,180,625 | -0.1 | -4,780 | +6.68 | +217,006 |
| Neonates that die | 10 | 334,498 | +2.2 | +71,934 | -6.9 | -224,572 |
| Maternal mortality | 0.5 | 15,296 | -0.12 | +875,673 | -18.7 | -6,692 |
| Maternal mortality per 100,000 live birth | 75 | | -19 | | 33 | |
| Neonatal mortality per 1,000 live births | 16 | | 3 | | -11 | |
| **Complications associated with increased CS rates** | | | | | | |
| Previa | 0.5 | 16,252 | -0.09 | -2,868 | +0.38 | +12,428 |
| Accreta | 0.4 | 13,384 | -0.06 | -1,912 | +0.38 | +12,428 |
| Hysterectomy | 0.3 | 10,516 | +8.97 | +291,572 | +0.24 | +7,648 |
| ICU admissions | 0.1 | 2,868 | -0.09 | -2,772 | +0.03 | +956 |
| Uterine Rupture | 0.4 | 117,585 | +3.73 | +15,296 | +3.4 | +4,780 |
| PPH requiring transfusion | 2.2 | 701,685 | +19.93 | +16,252 | +20 | +34,415 |
| **Complications associated with lack of access to CEmOC care** | | | | | | |
| Fistula | 0.003 | 956 | +0.03 | 0.0 | 0.0 | -860 |
| Incontinence | 0.009 | 2,868 | +0.08 | 0.0 | +0.1 | 0.0 |
| Stroke | 0.021 | 6,692 | +0.16 | -956 | +0.2 | 0.0 |
| Uterine Rupture | 0.4 | 117,585 | +3.73 | +15,296 | +3.4 | +4,780 |
| PPH requiring transfusion | 2.2 | 701,685 | +19.93 | +16,252 | +20 | +34,415 |

Mean number of complications per woman during her reproductive lifetime are presented as number of complications per 10,000 women and for the entire female population of reproductive age in India in 2018. Absolute numbers are presented for the baseline strategy [national average access to CEmOC and CS rate] and the difference between the baseline strategy and the rural and urban strategy are presented. A positive number represents an increase in the number of events while a negative value is a decrease in the number of events.

difference noted among rare complications such as stroke, which is likely due to their low prevalence.

Maternal mortality ratio (MMR) in India is estimated to be 145 maternal deaths per 100,000 live births, with only 46% of deaths occurring during labor, delivery, and 24 hours postpartum [32, 33]. Furthermore, 34% of maternal deaths in India are due to infection, hypertension, and abortion [34]. Since the model only accounts for deaths in the labor, delivery and immediate postpartum period with a focus on complications from hemorrhage and obstructed labor, the MMR in the model is appropriately lower than India's current value. Neonatal mortality rate (NMR) in India is estimated to be 23 deaths per 1,000 live births with 40% of deaths occurring during labor, delivery and first 24 hours [32, 35]. Since the model only accounts for deaths in the labor, delivery, and immediate postpartum periods, the NMR in the model is expectedly lower than India's current value.

## Interpretation of findings

Previous studies have looked at the cost-effectiveness of strategies to improve utilization and provision of maternal and newborn care in LMICs, generating strong evidence for the use of home-based newborn care with community health workers and traditional birth attendants, and adding services to routine antenatal care [36]. The present study is the first to look at the cost-effectiveness of strategies that combine access to CEmOC and CS rates to provide insight on strategies that could optimize the availability of obstetric surgical care.

Significant attention has been given to reducing CS rates, as the risk of maternal morbidity and mortality increases progressively with the number of CS each woman undergoes [7, 37]. However, most of these studies are from high-income countries and the morbidity and mortality consequences of increased CS rates have not been well characterized in resource limited settings. It is likely that the negative consequences of rising CS rates could be exacerbated in LMICs, due to higher fertility rates, and weaker surgical capacity to manage the increased rates of complications from abnormal placentation and repeat CS that result from increasing CS rates.

While it is important to minimize non-indicated CS, ensuring access to CEmOC is of paramount importance in efforts to decrease maternal and neonatal mortality and morbidity. Rural areas in India have lower rates of access to CEmOC with notable consequences. In one study, two thirds of maternal mortality occurred while women were seeking healthcare and three quarters of maternal deaths cluster in rural areas of the poorer states, despite these regions only representing one half of the live births [34]. Our results highlight the importance of access and how the strategy with the highest access to CEmOC, despite having higher than recommended CS rates, is cost-effective. A CEA comparing strategies for reducing maternal mortality in India found a cost effective strategy that involved increasing midwifes, obstetricians, and family planning and led to a prevention of 69% of mortality [24]. In concurrence with this result, our study demonstrates the importance of improving access to surgical care in CEmOC facilities. While in India, the rate of institutional deliveries has greatly increased, these do not necessarily include the surgical systems needed to perform CS.

## Clinical and policy implications

This study demonstrates that efforts to improve the provision of obstetric care in India must account for variation in access to CEmOC facilities and CS rates. Between these two factors, access to CEmOC facilities appears to drive cost-effectiveness among the competing strategies more than CS rate. Therefore, a focus on increasing access to obstetric surgical care in all settings, and particularly in underserved and rural areas, rural areas is warranted. This study did not assess strategies to scale up access to CS which would need to take into account training of workforce and new infrastructure. However, some strategies to increase workforce are already being implemented. In 2014, India began implementation of a program engaging surgeons to perform CS and manage obstetric complications to help address human resource limitations [38]. In concert with workforce, the scale up of CEmOC services must be accompanied by high quality of care, including appropriate use of CS when indicated.

Areas with high CS rates, particularly urban areas or the private sector, may benefit from interventions aimed at decreasing CS rates while preparing to deal with increasing co-morbidities following the trend in rising CS rates from the last decade. VBAC has been decreasing from 74.46% to 34.42% from 2005 to 2015 [39]. Increasing the number of women that have a TOLAC is an effective approach to decrease CS rates. TOLAC and elective repeat CS have been shown to be nearly equally cost-effective for the second delivery, and TOLAC becomes both less costly and more effective with subsequent deliveries [40]. Therefore, increasing

TOLAC remains a particularly important strategy in areas that are resource constrained and have high fertility rates.

## Limitations

Our model focused on the labor, delivery, and immediate post-partum periods as timepoints when access to surgical care is most critical. As a result, we excluded outcomes that could arise from abortions or the late postpartum period and long-term costs of maternal and neonatal mortality. Neonatal morbidity and other common causes of maternal mortality such as sepsis were not included in the analysis. The data for access to CEmOC facilities used in this model came from the Gujarat state, as no national level data is available. Fertility rates differ across India and our model did not include this variation. Within both urban and rural areas, there are variations in access to CS, and while CS may also be dependent upon income, education, private vs public sector, and other socioeconomic factors, these were not included in the analysis as we were limited by the paucity of data available. However, our methodology and model are robust and our sensitivity analysis and the findings on the outcomes of maternal and neonatal mortality are consistent with the current data from India.

## Conclusions

Preventable morbidity and mortality result from both lack of access to surgical obstetric care and high CS rates driven by non-indicated CS. This analysis on optimization of availability of CS demonstrates that the strategy with the highest access to CEmOC, despite having the highest CS rate, was cost-effective. The cost-effectiveness was driven by access to CEmOC rather than by CS rates. While efforts to decrease CS rates in areas of overutilization remains important, increasing access to CEmOC services is paramount to optimize outcomes. These findings could guide policymakers in India and other countries towards supporting increased access to CS while maintaining an appropriate CS rate and promoting equitable care for all women.

## Supporting information

**S1 Text. Appendix.** Fig A: (A): Markov states. (B): Decision tree for women with access to CEmOC care. (C): Decision tree for women who had no access to CEmOC care. Table A: Probabilities by number of CS. Fig B: Two-way sensitivity analysis of access to CEmOC facility and CS rates.
(DOCX)

## Author Contributions

**Conceptualization:** Lina Roa, Adeline A. Boatin, Mark Shrime.

**Data curation:** Lina Roa, Luke Caddell, Namit Choksi, Shylaja Devi, Adeline A. Boatin.

**Formal analysis:** Lina Roa, Jordan Pyda.

**Methodology:** Lina Roa, Namit Choksi, Jordan Pyda, Adeline A. Boatin, Mark Shrime.

**Software:** Mark Shrime.

**Supervision:** Adeline A. Boatin, Mark Shrime.

**Validation:** Lina Roa, Luke Caddell, Namit Choksi, Shylaja Devi.

**Visualization:** Lina Roa, Luke Caddell.

**Writing – original draft:** Lina Roa, Luke Caddell.

**Writing – review & editing:** Lina Roa, Luke Caddell, Namit Choksi, Shylaja Devi, Jordan Pyda, Adeline A. Boatin, Mark Shrime.

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
