## [Decision Letter · Decision Letter 0]

29 Mar 2022

PGPH-D-21-00715

Optimizing availability of obstetric surgical care in India: A cost-effectiveness analysis examining rates and access to Cesarean sections.

Dear Dr. Roa,

Thank you for submitting your manuscript to PLOS Global Public Health. After careful consideration, we feel that it has merit but does not fully meet PLOS Global Public Health’s publication criteria as it currently stands. Therefore, we invite you to submit a revised version of the manuscript that addresses the points raised during the review process.

We look forward to receiving your revised manuscript.

Kind regards,

Kathryn Chu, MD

Academic Editor

Journal Requirements:

1. In the Funding Information you indicated that no funding was received. Please revise the Funding Information field to reflect funding received.

Please ensure that the funders and grant numbers match between the Financial Disclosure field and the Funding Information tab in your submission form. Note that the funders must be provided in the same order in both places as well.

2. Please update the completed 'Competing Interests' statement. Please declare all competing interests beginning with the statement “I have read the journal's policy and the authors of this manuscript have the following competing interests:”.

3. Please provide separate figure files in .tif or .eps format only and remove any figures embedded in your manuscript file. Please ensure that all files are under our size limit of 20MB.

4. Please provide us with a direct link to the base layer of the map used in Figure 1 and ensure this location is also included in the figure legend. 

Please note that, because all PLOS articles are published under a CC BY license (creativecommons.org/licenses/by/4.0/), we cannot publish proprietary maps such as Google Maps, Mapquest or other copyrighted maps. If your map was obtained from a copyrighted source please amend the figure so that the base map used is from an openly available source.

Please note that only the following CC BY licences are compatible with PLOS licence: CC BY 4.0, CC BY 2.0  and CC BY 3.0, meanwhile such licences as CC BY-ND 3.0 and others are not compatible due to additional restrictions. If you are unsure whether you can use a map or not, please do reach out and we will be able to help you. 

The following websites are good examples of where you can source open access or public domain maps:

Additional Editor Comments (if provided):

Reviewers' comments:

Reviewer's Responses to Questions

**Comments to the Author**

1. Does this manuscript meet PLOS Global Public Health’s publication criteria? Is the manuscript technically sound, and do the data support the conclusions? The manuscript must describe methodologically and ethically rigorous research with conclusions that are appropriately drawn based on the data presented.

Reviewer #1: Partly

Reviewer #2: Yes

2. Has the statistical analysis been performed appropriately and rigorously?

Reviewer #1: Yes

Reviewer #2: Yes

3. Have the authors made all data underlying the findings in their manuscript fully available (please refer to the Data Availability Statement at the start of the manuscript PDF file)?

Reviewer #1: Yes

Reviewer #2: Yes

4. Is the manuscript presented in an intelligible fashion and written in standard English?

Reviewer #1: Yes

Reviewer #2: Yes

5. Review Comments to the Author

Reviewer #1: The research was rigorously done, however it was too technical and not easy to understand by a non-medical reader.

In my perspective, the “strategies” used namely A, B and C where not adequately explained and very confusing. Was the

national, rural and urban data used refer to as the “Strategies”, if yes, how are they strategies?

It was also unclear as to how Strategy C was still cost-effective, notwithstanding the increased number in maternal co-

morbidities and mortalities with the hypothesis stated, whereas in Strategy B (rural data), for instance with increased rates

of neonatal mortality and stillbirth, it was not registered as not cost-effective.

The term “cost-effectiveness” was not adequately contextually defined in the work as well as the term “CEmOC”. CEmOC

should clearly be define and indicates what packages or criteria of services delivery were available contextually in making

the “CEmOC”. Was it just by assumption of a standard (WHO Policy) of what a CEomC facility should be? or its just

for the assumption that the facility is located in an urban area? This should be clearly indicated at the introduction as this

would serve as an indicator for readers.

Reviewer #2: I commend the authors on this innovative study. I read it with interest. I have a few questions I'd appreciate their response on:

1. Is it common to use GDP per capita for the willingness to pay (WTP) threshold?

2. It's a little hard to follow the different strategies. Maybe just call them Rural Strategy, Urban Strategy, and Average Strategy?

3. It might be helpful to mention that extrapolation of limited data means that small variations end up being magnified -- this may be why when 55.7% of the population ends up having access to surgical care, it reportedly results in MORE complications "in the next 34 years." Is 34 years the difference between 14 year old and 49 year old? Because I'm not sure 34 is the right phrase you want there.

6. PLOS authors have the option to publish the peer review history of their article (what does this mean?). If published, this will include your full peer review and any attached files.

**Do you want your identity to be public for this peer review?** For information about this choice, including consent withdrawal, please see our Privacy Policy.

Reviewer #1: No

Reviewer #2: No

---

## [Decision Letter · Decision Letter 1]

15 Nov 2022

Optimizing availability of obstetric surgical care in India: A cost-effectiveness analysis examining rates and access to Cesarean sections.

PGPH-D-21-00715R1

Dear Dr Roa,

We are pleased to inform you that your manuscript 'Optimizing availability of obstetric surgical care in India: A cost-effectiveness analysis examining rates and access to Cesarean sections.' has been provisionally accepted for publication in PLOS Global Public Health.

Best regards,

Alice Zwerling, PhD

Academic Editor

Reviewer Comments (if any, and for reference):

Reviewer's Responses to Questions

**Comments to the Author**

1. If the authors have adequately addressed your comments raised in a previous round of review and you feel that this manuscript is now acceptable for publication, you may indicate that here to bypass the “Comments to the Author” section, enter your conflict of interest statement in the “Confidential to Editor” section, and submit your "Accept" recommendation.

Reviewer #2: All comments have been addressed

2. Does this manuscript meet PLOS Global Public Health’s publication criteria? Is the manuscript technically sound, and do the data support the conclusions? The manuscript must describe methodologically and ethically rigorous research with conclusions that are appropriately drawn based on the data presented.

Reviewer #2: Yes

3. Has the statistical analysis been performed appropriately and rigorously?

Reviewer #2: Yes

4. Have the authors made all data underlying the findings in their manuscript fully available (please refer to the Data Availability Statement at the start of the manuscript PDF file)?

Reviewer #2: Yes

5. Is the manuscript presented in an intelligible fashion and written in standard English?

Reviewer #2: Yes

6. Review Comments to the Author

Reviewer #2: All comments have been addressed. I commend the authors on this interesting and thoughtful paper.

7. PLOS authors have the option to publish the peer review history of their article (what does this mean?). If published, this will include your full peer review and any attached files.

**Do you want your identity to be public for this peer review?** For information about this choice, including consent withdrawal, please see our Privacy Policy.

Reviewer #2: No
